# Antibacterial and Antiallergic Effects of Three Tea Extracts on Histamine-Induced Dermatitis

**DOI:** 10.3390/ph17091181

**Published:** 2024-09-07

**Authors:** Zeting Huang, Lanyue Zhang, Jie Xuan, Tiantian Zhao, Weihua Peng

**Affiliations:** 1Guangzhou Zhongzhuang Meiye Cosmetics Co., Ltd., Guangzhou 510006, China; huangzeting@hanhoo.com (Z.H.); xuanjie@hanhoo.com (J.X.); 2Guangdong Provincial Key Laboratory of Plant Resources Biorefinery, School of Biomedical and Pharmaceutical Sciences, Guangdong University of Technology, Guangzhou 510006, China; 3Key Laboratory of Functional Foods, Guangdong Key Laboratory of Agricultural Products Processing, Sericulture & Agri-food Research Institute, Guangdong Academy of Agricultural Sciences, Ministry of Agriculture and Rural Affairs, Guangzhou 510610, China; lybluemoon@163.com; 4Department of Food Science, Rutgers University, New Brunswick, NJ 08901, USA

**Keywords:** tea extracts, dermatitis, antiallergic activity, skin barrier repairing

## Abstract

Atopic dermatitis (AD) is a persistent and recurrent inflammatory skin condition with a genetic basis. However, the fundamental reasons and mechanisms behind this phenomenon remain incompletely understood. While tea extracts are known to reduce histamine-induced skin allergies and inflammation, the specific mechanisms by which various types of Chinese tea provide their protective effects are still not fully elucidated. In this study, a model of skin itching induced by histamine is used to explore the functions and mechanisms of three types of tea extract (Keemun black tea (HC), Hangzhou green tea (LC), and Fujian white tea (BC)) in alleviating histamine-induced dermatitis. The components of three tea extracts are identified by UPLC-Q-TOF-MS, and we found that their main components are alkaloids, fatty acyls, flavonoids, organic acids, and phenols. The inhibitory effects of three types of tea extract on *Escherichia coli* (*E. coli*) and *Staphylococcus aureus* (*S. aureus*) in skin injury are investigated by MIC and flow cytometry. The three types of tea extract have an inhibitory effect on the growth of bacterial flora, with HC showing the best inhibitory activity. The effect of the three types of tea extract on histamine-induced dermatitis is also evaluated. Furthermore, itchy skin experiments, HE staining, toluidine blue staining, and immunohistochemical staining of mouse skin tissues were performed to determine the variations of scratching, epidermal thickness, mast cell number, IL-1β, and NGF content after the administration of the tea extracts. The three types of tea extracts all alleviate and inhibit skin itching, epidermal hyperplasia, and allergic dermatitis. BC effectively alleviates epidermal hyperplasia caused by skin allergies, and LC significantly downregulates NGF. HC reduces histamine-induced mast cell infiltration and downregulates IL-1β to alleviate skin itching. Consequently, tea emerges a potent natural product that can inhibit the growth of skin wound bacterial flora and exhibit skin repair effects on histamine-induced allergic dermatitis.

## 1. Introduction

Atopic dermatitis (AD) is a chronic inflammatory skin condition marked by dryness, eczema-like rashes, and intense itching. It often co-occurs with allergies, with up to 80% of sufferers having a familial history of allergic conditions [1]. Skin allergy, or hypersensitivity, involves abnormal immune responses driven by certain antigens, leading to tissue damage and physiological dysfunction. Common allergic skin diseases include eczema, urticaria, psoriasis, and atopic dermatitis. Type I hypersensitivity is particularly common in skin allergic reactions, characterized by histamine-mediated sensitization and pruritus [2,3]. With increasing environmental changes and cosmetic use, the incidence of atopic dermatitis is rising globally [4]. Although natural anti-allergic remedies such as green tea and Artemisia annua tea show promise, effective treatments are still limited [5,6]. Therefore, the development of safe and effective natural products for skin allergies remains a global priority.

Tea originates in China and is one of the three largest beverages in the world, with a long history of medicinal use [7]. It is classified into six fermentation categories: green tea (non-fermented), yellow tea (microfermented), white tea (slightly fermented), oolong tea (semi-fermented), black tea (fully fermented), and dark tea (post-fermented). These six types of tea belong to *Camellia sinensis* (L.) Kuntze; green tea does not require fermentation but is processed from fresh leaves through processes such as withering, rolling, and drying, thus maximizing the retention of nutrients such as tea polyphenols and vitamins. Black tea is produced via a process in which fresh leaves undergo withering, rolling, fermentation, and drying, resulting in a series of chemical changes in the chemical composition of the leaves, ultimately forming the unique qualities and characteristics of black tea [8]. Fermentation is an important process in the formation of high-quality black tea. The production method used for white tea is unique, as it is directly withered and dried without frying or kneading. Studies show that tea has strong anti-allergic effects, largely due to catechins, which are polyphenolic compounds with significant antioxidant properties [9,10,11,12]. Polyphenols can disrupt lipid–protein interfaces, inhibit biofilm formation, and have antimicrobial effects [13,14,15]. For example, a study evaluating the antibacterial activity of aqueous extracts from black tea and green tea against *Mutans streptococci* in saliva shows that both types of tea exhibit significant antibacterial effects [16]. Wahran M. Saod et al. successfully synthesized manganese oxide nanoparticles (MnO NPs) using green tea extract as a reducing agent. MnO NPs exhibit strong antibacterial activity against pathogenic bacteria such as *Escherichia coli*, *Klebsiella pneumoniae*, and *Pseudomonas aeruginosa* [17]. Moreover, skin allergies are often accompanied by bacterial infections or microbial imbalances, leading to the exacerbation of inflammation. Therefore, microbiological analysis helps us to understand the antibacterial effects of tea extracts and their impact on the skin microbial environment, and better understand the role of these extracts in anti-allergic reactions, providing data to support the development of natural therapeutic products. Catechins also provide whitening, moisturizing, anti-inflammatory, and nourishing benefits [18]. They inhibit the NF-κB signaling pathway and modulate inflammatory responses by affecting macrophages [19]. In addition, tea extract may help relieve skin symptoms such as redness and itching associated with allergies by lowering IgE and histamine levels, as well as inhibiting related transcription factors, suggesting its anti-inflammatory and anti-allergic effects [20].

Due to their different fermentation levels, the biochemical components of various teas are also different, so it is crucial to systematically evaluate their anti-allergic and antibacterial activities. This study investigates the effects of three types of tea extracts on histamine-induced skin allergies and their antibacterial properties, and analyzes the relationship between their active ingredients and related indicators. This study reveals differences in the teas’ efficacy and mechanisms, providing insights for the development of natural therapies with therapeutic and antibacterial properties for skin allergies.

## 2. Results

### 2.1. Components of Tea Extracts

Figure 1 is a complete scan of the positive and negative ions in Keemun black tea, Hangzhou green tea, and Fujian white tea extracted with water. The exact molecular weight test values of the detected compound match the theoretical values, and were then input into the mz-Cloud database of Compound Discoverer, analyzed, and identified.

As can be seen from Figure 1, the compositions of LC, HC, and BC are approximately the same. However, HC’s compositions are the most diverse. Appendix A and Figure 2 show the specific components of the Keemun black tea, Hangzhou green tea, and Fujian white tea extracts obtained during the final analysis. Among them, the main components of LC are caffeine, 1-stearoylglycerol, epigallocatechin gallate, (-)-gallocatechin, and 2,2′-methylenebis(4-methyl-6-tert-butylphenol). The main components of HC are caffeine, 1-stearoylglycerol, 2,2′-methylenebis(4-methyl-6-tert-butylphenol), D-(-)-quinic acid, and stearamide. The main components of BC are epigallocatechin gallate, caffeine, 1-stearoylglycerol, (-)-gallocatechin, and D-(-)-quinic acid. According to the literature, the main elements, caffeine, α,α-trehalose, erucamide, epigallocatechin gallate, choline, (-)-gallocatechin, gluconic acid, rutin, citric acid, and theobromine, have anti-inflammatory effects.

### 2.2. Minimum Inhibitory Concentration (MIC)

The MIC results for HC, LC, and BC against *E. coli* and *S. aureus* are shown in Figure 3. According to Figure 3A, compared to the positive drug tetracycline (TC), the MICs of the three tea extracts against *E. coli* are equal to or even higher than that of the positive drug (2.81 mg/mL), with MICs of 2.81 mg/mL for HC, 3.74 mg/mL for LC, and 4.21 mg/mL for BC. Among the three tea extracts, HC, which contains more caffeine, effectively inhibits the growth of *E. coli*. HC appears to inhibit *E. coli* most effectively, followed by LC. Figure 3B shows that black tea and white tea have similar and better effects on *S. aureus*, with MICs of 2.73 mg/mL for black tea and 3.03 mg/mL for white tea. The MIC of the positive drug against *S. aureus* is 2.77 mg/mL. Additionally, the MICs of LC and BC against *S. aureus* are higher than the positive TC (2.77 mg/mL). Organic acids inhibit *S. aureus*, and LC contains the least amount of organic acids compared to the other two tea extracts. Thus, LC has a worse bacteriostatic effect on *S. aureus* (5.85 mg/mL) than black tea and white tea. Therefore, HC shows superior inhibitory effects on both *E. coli* and *S. aureus* compared to the other two tea extracts.

### 2.3. Antibacterial Capability of Tea Extract by Flow Cytometry

Flow cytometry can be used to stain cells or bacteria with fluorescent dyes to analyze various cell characteristic parameters such as cell apoptosis and cycle, membrane potential, DNA fragments, and other data. In this experiment, live and dead bacteria are stained with SYTO-9/PI dye, and the number and content of live and dead bacteria are analyzed by the fluorescence intensity of the staining. As shown in the experimental results in Figure 4, after treatment with HC, LC, and BC, the content of dead *S. aureus* (Q1) increases compared to the control group, and the content of dead bacteria increases with the increase in tea concentration. Under the same concentration comparison, HC shows a better antibacterial effect on *S. aureus* than BC, while LC shows a worse antibacterial effect than HC and BC. These findings are somewhat similar to those in the existing literature. For example, Alghamdi et al. [21] find that green tea leaf extract exhibits significant antibacterial activity against *Bacillus subtilis*, especially ethanol extract, which may be related to the active ingredients it contains. In addition, Khan et al. [22] find that saponins extracted from green tea seeds have significant antibacterial effects on *E. coli*, *S. aureus*, and various *Salmonella*, and effectively reduce the level of pathogen infection in vivo, which is consistent with our findings on tea extracts.

### 2.4. Number of Scratching Times

Skin itching is associated with the release of histamine or related mediators from mast cells. In this experiment, we evaluate the inhibitory effect index of drugs on skin itching by establishing a histamine-induced acute itching model in mice after applying different drug groups and counting the number of scratches in the different groups of mice within 30 min. In Figure 5, the number of rat bites indicates that, compared with the control group, the allergic reaction induced by histamine injection results in a significant increase in the number of mice scratching due to skin itching. The treatments of the positive group, HC group, and BC group reduced the number of mice scratching compared with the model group, and it is statistically significant (*p* < 0.01). The LC treatment also had the effect of reducing the number of bites in mice, but its effect on relieving itching is slightly lower than the other groups. HC’s impact is comparable to that of a positive injection of diphenhydramine hydrochloride, effectively alleviating itchy skin associated with skin allergies.

### 2.5. Number of Hair Follicles

Epidermal hyperplasia, a common feature of skin damage from allergic dermatitis, is frequently used as a measure to evaluate the effectiveness of treatments in reducing this condition [21]. A histological analysis of the test skin, illustrated in Figure 6, reveals that epidermal hyperplasia is reduced in the positive group (DPH), HC group, LC group, and BC group compared to the model group. This suggests that applying the three tea extracts locally effectively inhibits epidermal hyperplasia induced by skin allergies. The skin thickness in Figure 6 shows that allergic reactions caused by histamine injection significantly increase epidermal thickness compared to the control group. Compared to the model group, treatments with positive controls, black tea, green tea, and white tea all result in a reduction in skin epidermal thickness. However, the effectiveness of black tea in reducing epidermal thickness is somewhat less pronounced than that of other treatments. White tea’s effect is comparable to that of a positive injection of diphenhydramine hydrochloride, effectively inhibiting epidermal hyperplasia caused by skin allergies.

### 2.6. Infiltration of Mast Cells

A notable feature of histamine-induced atopic dermatitis is the proliferation and infiltration of mast cells, whose cytoplasm contains basophilic granules that are stained bluish-purple with toluidine blue. Mast cell threshing can aggravate skin allergy symptoms. Therefore, toluidine blue staining is used to analyze the effects of HC, LC, and BC on mast cell proliferation and infiltration. From Figure 7, the number of mast cells in the model group is significantly higher than in the control group after histamine injection. Compared with the model group, the number of mast cells in the dermis of black tea, green tea, and white tea is significantly reduced, and the effect is better than that of the positive group (*p* < 0.01). Compared with other components, the proliferation and infiltration of mast cells in the skin tissue of mice in black tea extract is the lowest. Black tea effectively inhibits mast cell infiltration and granule removal and has an inhibitory effect on skin inflammation.

### 2.7. Immunohistochemical

#### 2.7.1. Expression of IL-1β in Mouse Dorsal Skin

The production of interleukin (IL-1β) and other inflammatory cytokines is crucial in skin inflammation. Histamine-induced allergic itching significantly increases the expression of genes related to various cytokines in the affected skin and significantly increases serum levels of total IgE. This results in heightened levels of several pro-inflammatory cytokines, including IL-4, IL-23, IL-33, IL-27, IL-1β, and TNF-α.

As depicted in Figure 8, the level of the pro-inflammatory cytokine IL-1β is notably higher in the model compared to the control. After treatment with the three tea extract solutions, the expression of IL-1β in the HC, BC, and LC groups is significantly decreased compared with the model group (*p* < 0.05, *p* < 0.01). The inhibition of IL-1β expression in black tea and green tea is significantly higher than that in the positive group (*p* < 0.01), with black tea showing the best performance.

#### 2.7.2. Expression of NGF in Mouse Dorsal Skin

Nerve growth factor (NGF) is a key target because it triggers acute neural sensitization and promotes the proliferation of nerve endings in the epidermis. Anti-NGF treatments can alleviate itching and allergic dermatitis. Figure 9 illustrates that, following modeling, NGF expression in the model group is significantly increased compared to the control group. After treatment with the tea extract solution, NGF expression decreases significantly compared with the model group, indicating that tea extract has a certain effect on skin barrier protection and repair. The downregulated expression of NGF in the BC group and HC extract group is close to that in the positive group (*p* < 0.01). The expression of NGF in the LC extract treatment group decreases most significantly and is better than that in the positive group, with statistical significance (*p* < 0.01). It has been proved that epigallocatechin gallate (EGCG) effectively alleviates symptoms and associated inflammatory markers of allergic rhinitis in a mouse model induced by ovalbumin [23]. This indicates that EGCG has a potential value for application in anti-allergic reactions. Overall, our study validates the broad efficacy of green tea and its derivatives in antibacterial and anti-allergic effects, similar to findings in the existing literature.

## 3. Discussion

Tea has a long history in China and shows many pharmacological effects [24]. Tea can be divided into green tea, white tea, oolong tea, yellow tea, black tea, and dark tea according to the degree of fermentation. These different types of tea show excellent anti-inflammatory and antioxidant effects, which attract the interest of researchers and can be used as new drugs for treatment [25,26]. Studies show that tea extracts have excellent antibacterial activity. Liu et al. find that four tea extracts, including green tea, oolong tea, black tea, and Fuzhuan tea, effectively inhibit the growth of *Enterococcus faecalis*, *Staphylococcus aureus*, *Escherichia coli*, and *Salmonella typhimurium* [27]. Our research also confirms that tea with different degrees of fermentation has different antibacterial effects, and black tea has better antibacterial effects on *E. coli* and *S. aureus* than white tea and green tea. In animal experiments, our study confirms that tea extract alleviates histamine-induced skin inflammation by inhibiting epidermal proliferation, reducing mast cell infiltration, and downregulating the expression of IL-1β and NGF, with a therapeutic effect that is even better than the positive drug diphenhydramine. Wang [28] confirms that tea extract downregulates the expression of inflammatory factors such as TNF-α, IL-1β, and IL-6, and improves skin redness and wrinkles. Therefore, the three tea extracts in this study alleviate histamine-induced skin inflammation and have certain antibacterial and anti-allergic effects, filling the gap in the current reports on the treatment of histamine-induced skin inflammation. They can be used as drugs for the treatment of skin inflammation, but their specific mechanisms of action need further clarification. This will be the focus of our future research.

## 4. Materials and Methods

### 4.1. Materials and Chemicals

Fujian white tea was collected from Guanyang Town, Fuding City, Fujian Province in March 2022; Hangzhou green tea was collected from Xihu District, Hangzhou City, Zhejiang Province in April 2022; and Keemun black tea was collected from Keemun County, Anhui Province in June 2022. Hangzhou green tea fresh leaves were inactivated with enzymes at 280 °C, kneaded for 45 min, and dried, while Fujian white tea fresh leaves were withered indoors for 48 h before drying. Keemun black tea was produced by first withering the fresh leaves indoors for 12 h, then kneading them for 45 min, followed by fermentation for 8 h, and finally drying the leaves. The extract was stored in the Institute of Nature Medicine & Green Chemistry (Guangdong University of Technology, Guangzhou, China) as a voucher specimen (no. ZLY-20220903-007, no. ZLY-20220903-008, no. ZLY-20220903-010). Ethanol, histamine, propylene glycol, and xylene were purchased from Aladdin Chemical Reagent Co., Ltd., Shanghai, China, all staining kits were purchased from Jiangsu Sumike Biotechnology Co., Ltd., Nanjing, China, and diphenhydramine was purchased from Sanjiu Pharmaceutical Co., Ltd., Shenzhen, China.

### 4.2. Extraction

Ten grams of tea powder and 100 mL of water were added to round-bottom flasks as solvent, respectively, and the extraction time was 40 min with the extraction temperature set at 90 °C [16]. The tea water extract was filtered after cooling, and the filtrate was combined. The filtrate was then placed in a −80 °C refrigerator to freeze; once the tea was frozen solid, it was transferred to a freeze dryer to be freeze-dried into tea powder, and the corresponding yield was calculated based on the weight of the extract. The HC yield was 29.5%, the BC yield was 25.8%, and the LC yield was 27.9%.

### 4.3. Components of Three Tea Extracts Determined by UPLC-Q-TOF-MS

In this study, water extracts from three tea types were analyzed using UPLC-Q-TOF-MS. The analysis was carried out on a UPLC-Q-TOF-MS platform (Thermo, Ultimate 3000 LC, Q Exactive HF). The column used was a C18 Zorbax Eclipse (1.8 μm × 2.1 × 100 mm). The chromatographic conditions were as follows: the column temperature was set to 30 °C; the flow rate was 0.8 mL/min; mobile phase A consisted of water with 0.1% formic acid, and B was pure acetonitrile; the injection volume was 10 μL; and the autosampler temperature was maintained at 4 °C. For ionization, the positive and negative modes had a heater temperature of 325 °C, sheath gas flow rate of 45 arb, auxiliary gas flow rate of 15 arb, purge gas flow rate of 1 arb, electrospray voltage of 3.5 kV, and capillary temperature of 330 °C. The S-Lens RF level was set at 55%. Scanning modes included first-order Full Scan (*m*/*z* 100–1500) and second-order data-dependent mass spectrometry (dd-MS2, TopN = 10). The resolution settings were 70,000 for primary mass spectrometry and 17,500 for secondary mass spectrometry. The collision mode used was High Energy Collision Dissociation (HCD).

### 4.4. Antibacterial Effect of Tea Extract Determined by Minimum Inhibitory Concentration (MIC)

It had been reported that the gut microbiota is closely related to allergic diseases such as intestinal inflammatory diseases, asthma, and histamine dermatitis. *E. coli* is the main bacterium in the human intestinal flora, and the proportion of *E. coli* in the intestinal tract of histamine dermatitis patients was higher than that in healthy individuals [19,20,29,30]. The diversity of the skin flora in histamine dermatitis patients is decreased compared to normal skin, mainly due to an increase in pathogenic *S. aureus* [31]. Therefore, *E. coli* and *S. aureus* were selected as antibacterial markers. Tetracycline (TC) and tea powder were prepared into 10, 5, 2.5, 1.25, and 0.625 mg/mL concentration samples. Each 96-well plate was added with 50 μL of the above concentrations of drugs, and then 50 μL of 10^6^–10^7^ CFU/mL *E. coli* and *S. aureus* bacterial solutions were added. Bacterial solutions containing *E. coli* and *S. aureus* were used as positive controls, respectively, and deionized water and drugs of different concentrations were used as negative controls [32]. Five duplicate values were set for each group. After incubation in a constant temperature incubator for 24 h, the OD value was measured at 620 nm with an enzyme marker, and the antibacterial rate was calculated according to the OD value. The concentration–antibacterial rate graph was made for the teas. Based on the line graph data and analysis with GraphPad Prism software version 8.0.2, the MIC value of tea with an antibacterial rate of 90% was determined as the tea concentration.
$$\mathrm{Antibacterial}\;\mathrm{rate}\%=1-\frac{\mathrm{experimental}\;\mathrm{group}\;\mathrm{OD}\;\mathrm{value}\;-\;\mathrm{negative}\;\mathrm{control}\;\mathrm{OD}\;\mathrm{value}}{\mathrm{positive}\;\mathrm{control}\;\mathrm{OD}\;\mathrm{value}\;-\mathrm{negative}\;\mathrm{control}\;\mathrm{OD}\;\mathrm{value}}\;\times 100\%$$

### 4.5. Antibacterial Effect of Tea Extract by Flow Cytometry

In this experiment, the SYTO 9/PI live bacteria/dead bacteria double staining kit was used to stain *S. aureus*, and the influence of different concentrations of the three tea extracts on bacterial activity was detected [33]. *S. aureus* suspensions were counted using cell count plates and diluted to 10^6^–10^7^ CFU/mL with deionized water. Then, 500 μL of the diluted bacterial solution was taken, and 500 μL of the aqueous solutions of tea water extracts with concentrations of of 5, 2.5, and 1.25 mg/mL were added to each solution, and cultured at 25~35 °C for 3 h. The bacterial solution was centrifuged with 0.9% sodium chloride solution at 5000 r/min for 5 min and washed 3 times. The volume of bacteria was suspended at 10^6^–10^7^ CFU/mL. Then, 3 μL of SYTO-9/PI dye solution (1.5 μL SYTO-9 dye solution and 1.5 μL PI dye solution, respectively) was added, and the solution was incubated at room temperature and away from light for 2 h, and then the bacterial solution was washed twice at 4000 r/min for 5 min with 0.9% sodium chloride solution, and resuspended to 10^6^–10^7^ CFU/mL. After passing through a 70 μm cell sieve, a flow cytometer was used for machine detection. Each sample was tested for 20,000 bacteria. Live and dead bacteria were detected based on scatter plot data from samples analyzed and processed by the software, and the content and proportions of dead and live bacteria were recorded.

### 4.6. Animals and Treatments

Experimental SPF-grade male KM mice (5 weeks old, 34–38 g weight, 60 mice) were purchased from Guangdong Laboratory Animal Center, Guangzhou, China. All experimental procedures followed institutional guidelines for the care and use of laboratory animals, with a focus on minimizing animal suffering throughout the study. The mice were kept at 22 °C in a 12 h light/dark cycle for 1 week and were randomly divided into the Control group: propylene glycol; positive control group: Diphenhydramine (DPH); model group: 1% Histamine; and HC group: 0.5% Keemun black tea; LC group: 0.5% Hangzhou green tea; and BC group: 0.5% Fujian white tea, with 10 mice in each group. The samples were deposited on the backs of the mice in an area of about 4 cm × 4 cm. From day 1 to day 7, the Model and Control groups received 200 μL of a 0.5% propylene glycol solution applied to the depilated area, while the Positive Control group was treated with 200 μL of a 0.5% propylene glycol hyaluronate solution. The HC, LC, and BC groups were treated with 200 μL of a propylene glycol solution containing the respective drug at 10 mg/mL. All the above groups were given their treatments once a day for 7 days. On day 7, a histamine-induced mouse model of acute pruritus was established. The positive group was given diphenhydramine hydrochloride dissolved in normal saline solution by intraperitoneal injection at a dose of 6 mg/mL and 200 μL/mouse. After the end of administration, 30 min later, an intraperitoneal injection of 100 μL of normal saline was given to the mice in the control group, while mice in the other groups were injected with 100 μL histamine by intraperitoneal injection (dissolved in normal saline solution at the concentration of 1 mg/mL). On the 7th day, the number of scratching times in 30 min was recorded, and the skin on the backs of the mice was photographed and its appearance was documented. The entire process of lifting and scraping the modeling site with the hind paw of a mouse until the hind paw landed was considered a scraping action. The mice were euthanized by intraperitoneal barbiturate injection with a dose of 125 mg/kg and a concentration of 5%. The animals were euthanized and dorsal skin tissues were fixed and frozen in paraformaldehyde for subsequent study. All animal experiments were approved by the Animal Experiment Ethics Committee of Guangdong University of Technology (approval number SCXK/2013–0002).

### 4.7. Hematoxylin and Eosin (HE) Staining of Hair Follicles

To assess hair follicles, the HE staining kit (g1120; Solarbio, Beijing, China) was used. At the end of day 7, skin samples (1 × 1 cm paraffin sections) from each histamine-induced acute pruritus mouse model group were collected and subjected to HE staining. The sections were dewaxed in xylene (20 min), alcohol (10 min), and distilled water (5 min), followed by staining with hematoxylin (3–8 min). The stained sections were then microscopically examined and analyzed with Image Pro Plus 6.0 software to determine the ratio of stained to unstained hair follicles.

### 4.8. Toluidine Blue Staining

The paraffin sections were immersed sequentially in xylene I for 20 min, xylene II for 20 min, absolute ethanol I for 5 min, absolute ethanol II for 5 min, and 75% alcohol for 5 min, followed by rinsing with tap water. Tissue sections were then stained with toluidine blue for 2–5 min, rinsed with water, differentiated using 0.1% glacial acetic acid, and washed again with water, with microscopic monitoring. The sections were dried in an oven, then cleared with xylene for 10 min and mounted with neutral glue. Finally, the samples were microscopically examined, and images were captured and analyzed.

### 4.9. Immunohistochemical Staining

Mouse skin samples (2 × 0.8 cm) were first rinsed to eliminate any residual blood, then dried and sliced for dewaxing. These slices were placed in a repair chamber containing a citric acid antigen retrieval buffer (pH = 6.0) and subjected to antigen retrieval in a microwave oven. After heating and cooling, the samples were washed with PBS (3 × 5 min) and treated with 3% hydrogen peroxide for 25 min in the dark. Following this, the samples were washed again with PBS and incubated with 3% BSA at room temperature for 30 min. After removing the blocking solution, primary antibodies were applied at 4 °C, followed by a PBS wash. The sections were then treated with secondary antibodies and incubated at room temperature in the dark for 50 min. After a final PBS wash, the samples were dehydrated and examined microscopically to assess NGF and IL-1β levels in the mouse skin.

### 4.10. Statistical Analysis

Data analysis was performed with GraphPad Prism version 8.0.2. The results were presented as mean ± standard deviation (X ± SD). To compare means across multiple groups, one-way analysis of variance (ANOVA) was used, while pairwise comparisons between smaller groups were assessed using two-sided Student’s *t*-tests. A *p*-value of less than 0.05 was considered statistically significant, and a *p*-value of less than 0.01 was considered highly significant.

## 5. Conclusions

In summary, our study demonstrates that the extracts of HC, LC, and BC possess remarkable antibacterial and anti-allergic properties. HC exhibits the most potent antibacterial effect against *E. coli* with an MIC of 2.81 mg/mL and is effective in reducing histamine-induced mast cell infiltration and down-regulating IL-1β expression. Its primary mechanism involves regulating oxidative stress and inflammatory factors, thereby alleviating skin pruritus. BC shows a superior performance in inhibiting epidermal hyperplasia and may act similarly to diphenhydramine hydrochloride by blocking histamine release. LC, although less effective in antibacterial assays, shows significant ability to downregulate nerve growth factor (NGF), potentially alleviating pain associated with skin damage and protecting against further skin injury. These findings suggest that these tea extracts could serve as promising natural anti-allergic agents. The difference in effectiveness observed among tea types underscores the need for tailored approaches depending on the specific skin condition. Future research should focus on elucidating the precise mechanisms by which these tea extracts exert their effects and exploring their potential in clinical settings.

## Figures and Tables

**Figure 1 pharmaceuticals-17-01181-f001:**
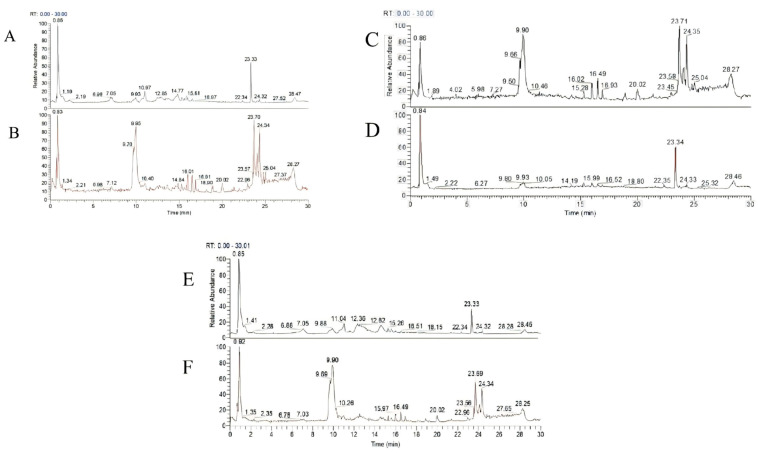
LC extract chromatograms in positive (**A**) and negative (**B**) modes. HC extract chromatograms in positive (**C**) and negative (**D**) modes. BC extract chromatograms in positive (**E**) and negative (**F**) modes.

**Figure 2 pharmaceuticals-17-01181-f002:**
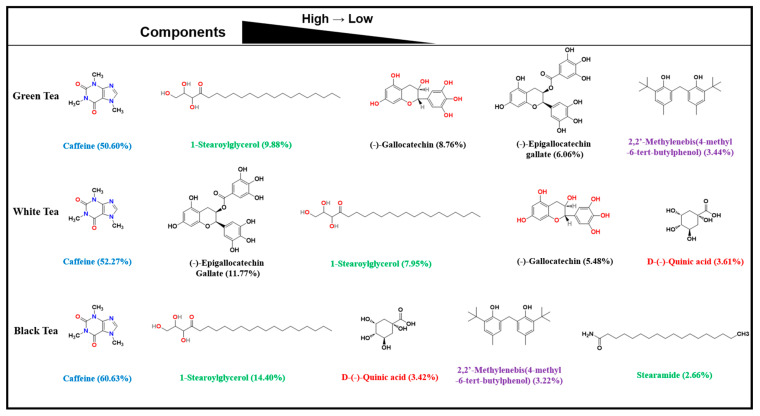
Chemical structures of main constituents of the three tea extracts.

**Figure 3 pharmaceuticals-17-01181-f003:**
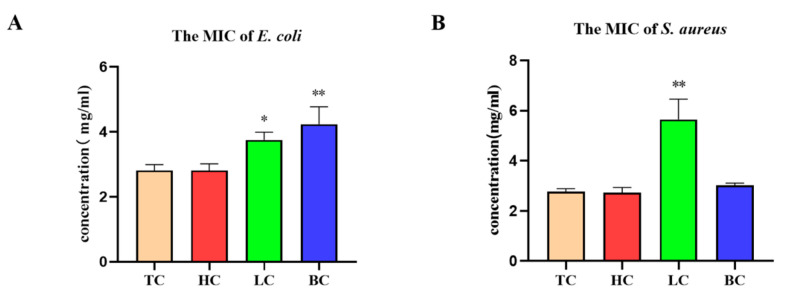
Inhibitory effect of three tea extracts on *E. coli* and *S. aureus*. MICs of three tea extracts on (**A**) *E. coli* and (**B**) *S. aureus*, * *p* < 0.05, ** *p* < 0.01.

**Figure 4 pharmaceuticals-17-01181-f004:**
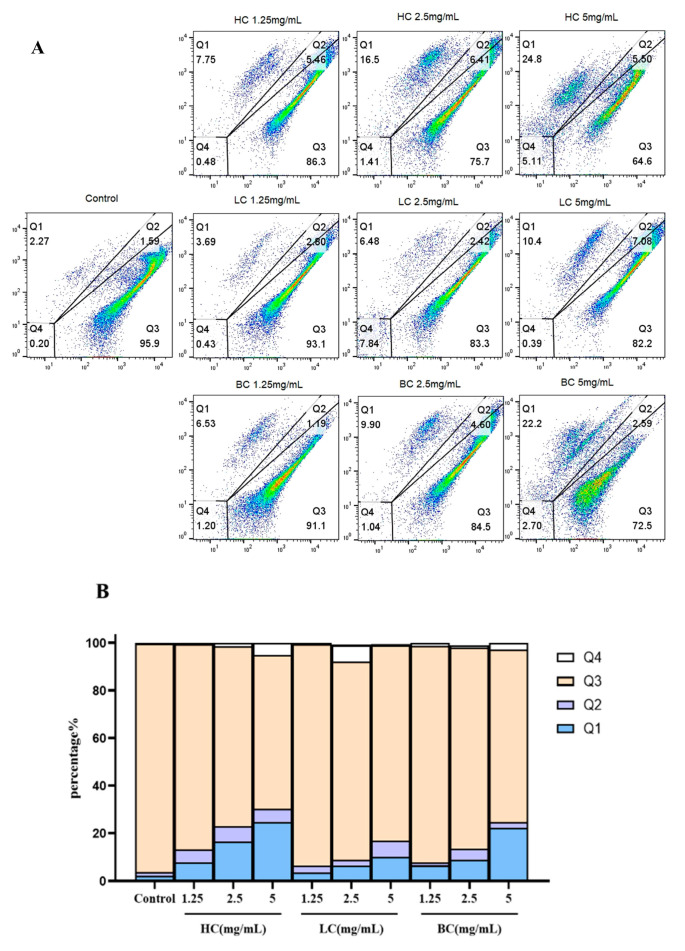
Antibacterial effects of three tea extracts on *S. aureus.* (**A**) Results of flow pattern of *S. aureus*. (**B**) Histogram results of *S. aureus* (areas Q1, Q2, Q3, and Q4 in this Figure are dead, damaged, alive, and unstained bacterial areas, respectively).

**Figure 5 pharmaceuticals-17-01181-f005:**
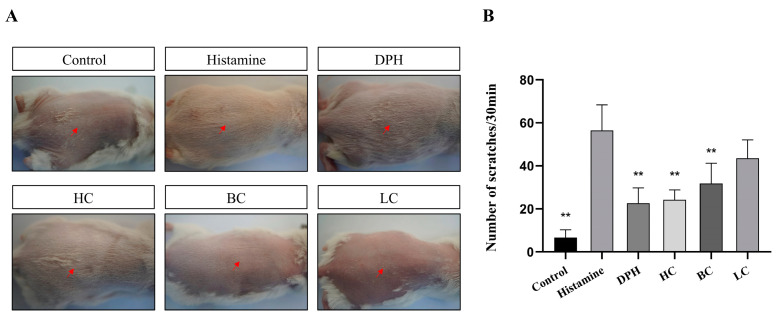
Number of scratches. (**A**) Pictures of mice backs. (**B**) Number of scratches (*n* = 10, compared with model group, ** *p* < 0.01). The red arrow represents the scratch area of the mouse.

**Figure 6 pharmaceuticals-17-01181-f006:**
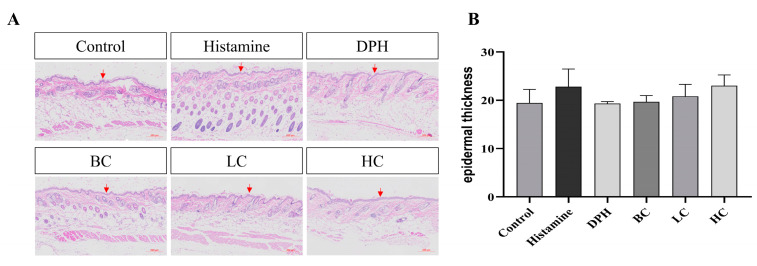
HE staining results, (**A**) pictures of staining for mice, and (**B**) epidermal thickness (*n* = 10). The red arrow represents the epidermis of mouse skin.

**Figure 7 pharmaceuticals-17-01181-f007:**
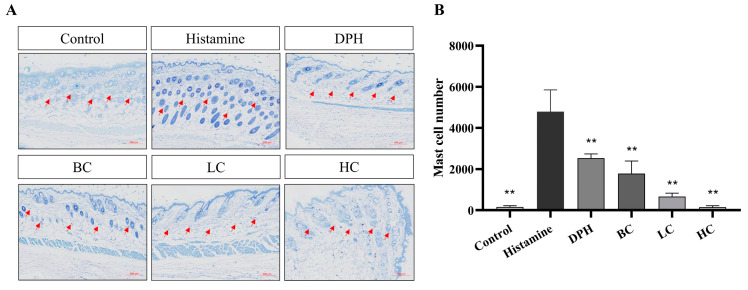
Toluidine blue staining results, (**A**) pictures of staining for mice, and (**B**) mast cell number (*n* = 10, compared with model group, ** *p* < 0.01). The red arrow indicates mast cells in the skin of mice.

**Figure 8 pharmaceuticals-17-01181-f008:**
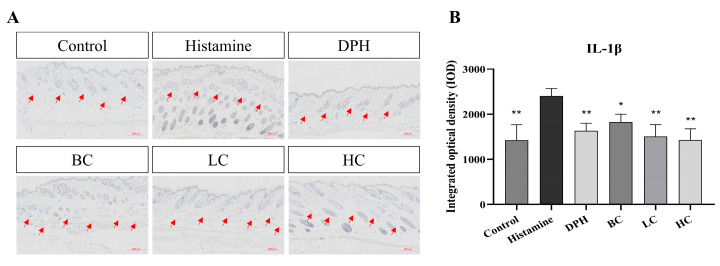
(**A**) Immunohistochemical test results for IL-1β. (**B**) Integrated optical density of IL-1β (*n* = 10, compared with model group, * *p* < 0.05, ** *p* < 0.01). The red arrow indicates the IL-1β cytokine in the mouse skin.

**Figure 9 pharmaceuticals-17-01181-f009:**
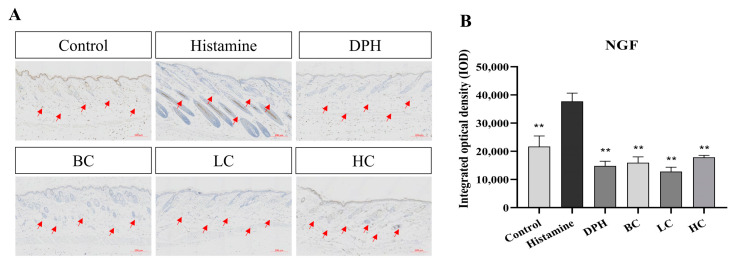
(**A**) Immunohistochemical test results for NGF. (**B**) Integrated optical density of NGF (*n* = 10, compared with model group, ** *p* < 0.01). The red arrow indicates the NGF cytokine in the mouse skin.

## Data Availability

The data presented in this study are available on request from the corresponding author. The data are not publicly available due to privacy or ethical reasons.

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
