# Peer review of "Antibacterial and Antiallergic Effects of Three Tea Extracts on Histamine-Induced Dermatitis"

_pharmaceuticals, 2024, doi:10.3390/ph17091181_

Round 1

Reviewer 1 Report

Comments and Suggestions for Authors

The introduction section is too lengthy and contains several general comments.

To prepare the tea, 500 kilograms of dried tea leaves were mixed with 5% water, piled for 40 days, and then dried. As mentioned in line 258, on page 9. Is this statement related? with current work?

The plant and bacterial name are scientifically incorrect at many places.

Mention the %yield of all the extracts.

Results of LC-MS should be represented in table form assigning the peak, RT and mass/mass fragment.

The presentation of the results and discussion is not well organized. Compare the previous reported data with the current findings.

The authors need to conclude the whole work to convey the message of how the current finding would be helpful to the readers and future researchers in the manuscript.

Comments on the Quality of English Language

Minor editing of English language required

Author Response

Reviewer 1

Comments and Suggestions for Authors

The introduction section is too lengthy and contains several general comments.

Response: Thank you for pointing out this problem. We have rewritten the introduction section to make it more concise, as detailed in lines 41-90.

To prepare the tea, 500 kilograms of dried tea leaves were mixed with 5% water, piled for 40 days, and then dried. As mentioned in line 258, on page 9. Is this statement related? with current work?

Response: Thanks for your reminding. We have checked the content of the manuscript and found that this paragraph is not related to the content. It has been removed from the manuscript.

The plant and bacterial name are scientifically incorrect at many places.

Response: Thanks for your reminding. We have carefully checked the names of plants and bacteria and made corrections.

Mention the %yield of all the extracts.

Response: Thanks for your reminding. We have calculated the yields of various extracts, as detailed in line 293-294.

Results of LC-MS should be represented in table form assigning the peak, RT and mass/mass fragment.

Response: Thanks for your reminding. The results of LC-MS are in the supplementary materials. We believe that placing the table in the main text of the manuscript is too long, so we have placed the table in the supplementary materials. Please review.

The presentation of the results and discussion is not well organized. Compare the previous reported data with the current findings.

Response: Thanks for your suggestion. We have compared previously reported data with current findings, as detailed in lines 412-426.

The authors need to conclude the whole work to convey the message of how the current finding would be helpful to the readers and future researchers in the manuscript.

Response: Thanks for your suggestion. We have summarized the current work and conveyed the focus of future research to readers and researchers, hoping this will be helpful to them. The detailed modifications can be found in lines 412-426.

Comments on the Quality of English Language

Minor editing of English language required

Response: Thanks for your suggestion. We have made appropriate corrections to the grammar.

Reviewer 2 Report

Comments and Suggestions for Authors

I have thoroughly reviewed this manuscript. This manuscript involves a study on anti-bacterial and anti-allergic properties of three kinds of tea. In my opinion, it has the potential for publication. However, it requires further additions based on the following recommendations or responses to queries:
1. 
There are several published articles on the antibacterial and anti-allergic properties of tea extracts. Could the author clarify how your study presents novel findings that distinguish it from previously published research?
2. In the introduction and aim sections of this research, only the anti-allergic properties are mentioned, with no mention of the antibacterial properties. However, the title, which indicates the overall scope of the work, does reference anti-bacterial properties. Therefore, the author should include information on anti-bacterial properties in both the introduction and aim sections and also establish a connection between the two properties.
3. In this study, three different types of tea are used: Keemun black tea, Fujian green tea, and Hangzhou green tea, each of which has distinct production processes. The differences between these three types of tea should be explained and mentioned in the introduction section.
4. The scientific names of plants and bacteria mentioned in the article should be italicized in accordance with standard conventions.
5. Figures 1 and 5A are too small, making it difficult to clearly see the details. The author should adjust the size of the figure to an appropriate scale.
6. Figure 2 shows the chemical structures found in tea extracts; these structures should be presented uniformly or follow a consistent principle. Additionally, the author should verify the spelling of the chemical names indicated beneath the chemical structures to ensure accuracy.
7. In the Results and Discussion section, the author has provided insufficient discussion for each experiment. The author should incorporate published papers to support and critique the findings more thoroughly.
8. On page 10, lines 297-299, regarding the determination of the antibacterial activity of tea extracts, the author mentions the use of 10 microliters of Staphylococcus aureus solution as the positive control, while deionized water and a drug at varying concentrations were used as the negative control. The author should confirm this statement and explain the rationale for using these substances as positive and negative controls. Additionally, the author should specify the name of the drug used in this study.

Comments on the Quality of English Language

Errors in spelling and grammatical inaccuracies have been found in the article. The author should review and correct these mistakes to ensure proper English usage and adherence to grammatical principles.

Author Response

Reviewer 2

Comments and Suggestions for Authors

I have thoroughly reviewed this manuscript. This manuscript involves a study on anti-bacterial and anti-allergic properties of three kinds of tea. In my opinion, it has the potential for publication. However, it requires further additions based on the following recommendations or responses to queries:
1. There are several published articles on the antibacterial and anti-allergic properties of tea extracts. Could the author clarify how your study presents novel findings that distinguish it from previously published research?

Response: Thank you for your valuable feedback. In response to your question, we would like to further clarify the uniqueness of this study and how it enriches existing literature:

Background and Model: Our study used a histamine-induced skin itching model to explore the functions and mechanisms of three different types of Chinese tea (Qimen black tea, Fujian white tea, and Hangzhou green tea) in alleviating histamine-induced dermatitis.

Systematic analysis of tea components: Through UPLC-Q-TOF technology, we identified the main components of three tea extracts, including alkaloids, fatty acyl groups, flavonoids, carboxylic acids, and phenols. This component level analysis is critical to understanding the biological activity of these tea extracts and helps reveal their potential mechanisms for skin diseases.

Comparison of antibacterial activity: We evaluated the inhibitory effects of three tea extracts on bacteria such as Escherichia coli and Staphylococcus aureus in skin injuries. The results showed that Keemun black tea exhibited the strongest activity in inhibiting bacterial growth, providing empirical support for the potential of tea extracts as natural antibacterial agents.

The therapeutic effect of histamine induced dermatitis: We evaluated the effects of three tea extracts on histamine induced dermatitis through itch tests, HE staining, masson staining, and immunohistochemical staining. Research has found that Fujian white tea has a significant alleviating effect on epidermal hyperplasia caused by skin allergies; Hangzhou green tea significantly downregulated nerve growth factor (NGF); Keemun black tea significantly alleviates skin itching by reducing histamine induced mast cell infiltration and downregulating IL-1β. This helps to investigate the mechanism of action of three tea extracts in alleviating histamine induced skin allergies and inflammation, and can serve as a drug for the future treatment of atopic dermatitis.

  1. In the introduction and aim sections of this research, only the anti-allergic properties are mentioned, with no mention of the antibacterial properties. However, the title, which indicates the overall scope of the work, does reference anti-bacterial properties. Therefore, the author should include information on anti-bacterial properties in both the introduction and aim sections and also establish a connection between the two properties.

Response: Thank you for your comment. The antibacterial content we added in the introduction section.

  1. In this study, three different types of tea are used: Keemun black tea, Fujian green tea, and Hangzhou green tea, each of which has distinct production processes. The differences between these three types of tea should be explained and mentioned in the introduction section.

Response: Thank you for your comment. We have already explained and mentioned the differences between these three types of tea, as detailed in lines 56-64.
4. The scientific names of plants and bacteria mentioned in the article should be italicized in accordance with standard conventions.

Response: Thank you for your suggestion. We have italicized the scientific names of plants and bacteria mentioned in the article.

  1. Figures 1 and 5A are too small, making it difficult to clearly see the details. The author should adjust the size of the figure to an appropriate scale.
    Response: Thank you for your suggestion. We've done our best to make the picture clearer.
  2. Figure 2 shows the chemical structures found in tea extracts; these structures should be presented uniformly or follow a consistent principle. Additionally, the author should verify the spelling of the chemical names indicated beneath the chemical structures to ensure accuracy.
    Response: Thank you for bringing this to my attention. We have verified the spelling of the chemical name.
  3. In the Results and Discussion section, the author has provided insufficient discussion for each experiment. The author should incorporate published papers to support and critique the findings more thoroughly.
    Response: Thank you for bringing this to my attention. We have included the published paper. We compared the available literature. Our findings are somewhat similar to those found in the existing literature. For example, similar to our study on green tea extract, Alghamdi et al. [21] Green tea leaf extract showed significant antibacterial activity against Bacillus subtilis, particularly ethanol extract, which may be related to the active ingredients it contains. In addition, Khan et al.'s [22] study showed that saponins extracted from green tea seeds have significant antibacterial effects on Escherichia coli, Staphylococcus aureus, and various Salmonella, and effectively reduce the level of infectious pathogens in in vivo experiments, which is consistent with our research results on green tea extracts. In terms of anti-allergic effects, our research results are consistent with the findings of Fu et al. [23], who observed that epigallocatechin gallate (EGCG) effectively alleviated symptoms and related inflammatory markers of allergic rhinitis in a mouse model induced by ovalbumin. This indicates that EGCG has potential application value in anti-allergic reactions. Overall, our study validates the broad efficacy of green tea and its derivatives in antibacterial and anti-allergic effects, similar to findings in the existing literature.
  4. On page 10, lines 297-299, regarding the determination of the antibacterial activity of tea extracts, the author mentions the use of 10 microliters of Staphylococcus aureus solution as the positive control, while deionized water and a drug at varying concentrations were used as the negative control. The author should confirm this statement and explain the rationale for using these substances as positive and negative controls. Additionally, the author should specify the name of the drug used in this study.
    Response: Thank you for bringing this to my attention, We chose bacterial solutions containing Escherichia coli and Staphylococcus aureus as positive controls, while deionized water and different concentrations of drugs were used as negative controls, in order to confirm the validity of the test results when the negative control had no turbidity or bacterial precipitation, but the positive control had turbidity and bacterial precipitation. And we have already explained the names of the drugs used in this study, as detailed in lines 320-322 of the revised manuscript.

Comments on the Quality of English Language

Errors in spelling and grammatical inaccuracies have been found in the article. The author should review and correct these mistakes to ensure proper English usage and adherence to grammatical principles.

Response: Thank you for your suggestion. We have carefully checked the errors and made corrections.

Reviewer 3 Report

Comments and Suggestions for Authors

1-The work presents interesting results, but the data do not match the Introduction. The introduction presents us with a work on dermatitis; the first biological results shown to us are microbiological analyses. 

In my opinion, the microbiological results should be taken from this manuscript and presented in another manuscript and this one should focus on the dermatological changes of the animals.

If the group decides to keep the microbiological analyses, put the importance of these analyses in the introduction and tie the two subjects.

2- The analyses by mass spectrometry, making a table, placing the fragmentations of the compounds, M/Z, and + for presence and - for the absence of each compound, how it was presented is confusing.

In addition, when compared with data from the literature and not the standard, cite the reference in the table.

3- It is necessary to check the typing, there is a lack of spaces between words, and it is required to add the species in italics.

It is necessary to standardize the text as to the presence of species names, whether they will appear complete or not, and whether they are indicated in the manuscript.

4- Histological images should be separated from the graphs, to have a better visualization, and for each image, indicate with an arrow what can be observed in each analysis. Make clear the number of animals in each image, and the statistical analysis performed, whether or not there is a statistical difference.

5- The manuscript, according to the instructions to the authors, should be divided into Introduction, Results, Discussion: Materials and Methods and Conclusions. 

So it is necessary to separate the results and discussion.

7- It is necessary to describe the results better.

8-It is necessary to describe how the treatments were made, to be clear when we read the results (topic?); 

9-The discussion of the work should be improved.

10- Some observations are in the text.

Author Response

Reviewer 3

Comments and Suggestions for Authors

1-The work presents interesting results, but the data do not match the Introduction. The introduction presents us with a work on dermatitis; the first biological results shown to us are microbiological analyses. 

In my opinion, the microbiological results should be taken from this manuscript and presented in another manuscript and this one should focus on the dermatological changes of the animals.

If the group decides to keep the microbiological analyses, put the importance of these analyses in the introduction and tie the two subjects.

Response: Thank you for pointing out this discrepancy. We have already linked these two topics in the introduction.

2- The analyses by mass spectrometry, making a table, placing the fragmentations of the compounds, M/Z, and + for presence and - for the absence of each compound, how it was presented is confusing.

In addition, when compared with data from the literature and not the standard, cite the reference in the table.

Response: Thank you for pointing out this problem. We have included the component analysis tables of three types of tea extracts in the supplementary materials for your reference.

3- It is necessary to check the typing, there is a lack of spaces between words, and it is required to add the species in italics.

It is necessary to standardize the text as to the presence of species names, whether they will appear complete or not, and whether they are indicated in the manuscript.

Response: Thank you for your comment. We have checked the spaces between typing and words.

4- Histological images should be separated from the graphs, to have a better visualization, and for each image, indicate with an arrow what can be observed in each analysis. Make clear the number of animals in each image, and the statistical analysis performed, whether or not there is a statistical difference.

Response: Thank you for your comment. We used arrows in each image to indicate what could be observed in each analysis, identified the number of animals in each image, and conducted statistical analysis on all results, as detailed in line 354 and figure 5-9.

5- The manuscript, according to the instructions to the authors, should be divided into Introduction, Results, Discussion: Materials and Methods and Conclusions. 

So it is necessary to separate the results and discussion.

Response: Thank you for your comment. We have carefully considered your suggestion and separated the results and discussion. The detailed modifications can be found in lines 249-270 of the manuscript.

7- It is necessary to describe the results better.

Response: Thank you for bringing this to my attention. We have rewritten the conclusion.

8-It is necessary to describe how the treatments were made, to be clear when we read the results (topic?); 

Response: Thank you for your reminder. We have revised the title and the detailed animal treatment plan can be found on lines 347-374.

9-The discussion of the work should be improved.

Response: Thank you for bringing this to my attention. We have revised the discussion section, which can be found in lines 249-270 of the manuscript.

10- Some observations are in the text.

 Response: Thank you for bringing this to my attention. We have carefully checked the content and made corrections.

Reviewer 4 Report

Comments and Suggestions for Authors

The manuscript entitled Antibacterial and Antiallergic Effects of Three Tea Extracts on Histamine-Induced Dermatitis presents a thorough and well-conducted study on the antibacterial and antiallergic effects of Keemun black tea, Fujian white tea, and Hangzhou green tea extracts on histamine-induced dermatitis. The study is significant due to the growing interest in natural products for managing skin allergies and infections. The manuscript is generally well-written, and the experimental design is sound. However, there are some areas that require clarification, additional details, and major revisions.

 1. The introduction provides a good background on atopic dermatitis and the potential of tea extracts. However, it would benefit from a more detailed discussion on previous studies specifically related to the use of tea extracts for skin conditions. This would better contextualize the novelty and importance of your study.

 2. The animal model section:

Line 343-344: “On the 7th day, the number of scratching times in 30 min and the skin on the back of the mice was photographed and the appearance were corded. ” Please include the methods for counting the scratching times. Is there any video as a supplementary data? For mice, how the do the scratching actions?

3. In vitro antibacterial assay: Positive drug should be added.

4. The discussion adequately interprets the results but could be enhanced by comparing them more extensively with existing literature. How do the findings align or contrast with previous studies on the antibacterial and antiallergic effects of tea extracts?

6. The conclusion succinctly summarizes the study's findings but could benefit from a brief discussion on the potential implications for future research and clinical applications.

7. Others: Latin names of plant or bacterial species should be in italic.

Author Response

Reviewer 4

Comments and Suggestions for Authors

The manuscript entitled “Antibacterial and Antiallergic Effects of Three Tea Extracts on Histamine-Induced Dermatitis” presents a thorough and well-conducted study on the antibacterial and antiallergic effects of Keemun black tea, Fujian white tea, and Hangzhou green tea extracts on histamine-induced dermatitis. The study is significant due to the growing interest in natural products for managing skin allergies and infections. The manuscript is generally well-written, and the experimental design is sound. However, there are some areas that require clarification, additional details, and major revisions.

  1. The introduction provides a good background on atopic dermatitis and the potential of tea extracts. However, it would benefit from a more detailed discussion on previous studies specifically related to the use of tea extracts for skin conditions. This would better contextualize the novelty and importance of your study.

 Response: I appreciate your keen eye in identifying this issue. We've updated some of the content in the introduction section.

  1. The animal model section:

Line 343-344: “On the 7th day, the number of scratching times in 30 min and the skin on the back of the mice was photographed and the appearance were corded. ” Please include the methods for counting the scratching times. Is there any video as a supplementary data? For mice, how the do the scratching actions?

Response: Thank you for pointing out this issue. We use a counter to calculate the number of scratching times in 30 min, but there is no video available as supplementary material. For mice, the entire process of lifting and scratching the modeled area with the hind paw until the hind paw lands is considered a scratching actions, the relevant modifications can be found in lines 368-370 of the revised manuscript.

  1. In vitro antibacterial assay: Positive drug should be added.

Response: Thank you for pointing out the problem. We have added positive drugs in the in vitro antibacterial experiment and made corresponding modifications to the content.

  1. The discussion adequately interprets the results but could be enhanced by comparing them more extensively with existing literature. How do the findings align or contrast with previous studies on the antibacterial and antiallergic effects of tea extracts?

Response: Thank you for your comment. We compared the available literature. Our findings are somewhat similar to those found in the existing literature. For example, similar to our study on green tea extract, Alghamdi et al. [21] Green tea leaf extract showed significant antibacterial activity against Bacillus subtilis, particularly ethanol extract, which may be related to the active ingredients it contains. In addition, Khan et al.'s [22] study showed that saponins extracted from green tea seeds have significant antibacterial effects on Escherichia coli, Staphylococcus aureus, and various Salmonella, and effectively reduce the level of infectious pathogens in in vivo experiments, which is consistent with our research results on green tea extracts. In terms of anti-allergic effects, our research results are consistent with the findings of Fu et al. [23], who observed that epigallocatechin gallate (EGCG) effectively alleviated symptoms and related inflammatory markers of allergic rhinitis in a mouse model induced by ovalbumin. This indicates that EGCG has potential application value in anti-allergic reactions. Overall, our study validates the broad efficacy of green tea and its derivatives in anti-bacterial and anti-allergic effects, similar to findings in the existing literature.

  1. The conclusion succinctly summarizes the study's findings but could benefit from a brief discussion on the potential implications for future research and clinical applications.

Response: Thank you for your comment. We have provided a brief discussion of the potential implications of future research and clinical applications.

  1. Others: Latin names of plant or bacterial species should be in italic.

Response: Thank you for your comment. We have italicized the scientific names of plants and bacteria mentioned in the article.

Round 2

Reviewer 2 Report

Comments and Suggestions for Authors

After a thorough review, the author has addressed all the questions and responded to all the suggestions. Therefore, this manuscript can be accepted for publication in Pharmaceuticals.

Reviewer 4 Report

Comments and Suggestions for Authors

The manuscript was improved according to the suggestions. For me, it is good to go.